

# Spatiotemporal gait characteristics post-total hip arthroplasty and its impact on locomotive syndrome: a before-after comparative study in hip osteoarthritis patients

Shigeaki Miyazaki[1], Yoshinori Fujii[2], Kurumi Tsuruta[3], Saori Yoshinaga[4], Amy Hombu[5], Taro Funamoto[6], Takero Sakamoto[6], Takuya Tajima[6], Hideki Arakawa[1], Tsubasa Kawaguchi[1], Jun Nakatake[1] and Etsuo Chosa[6]

[1] Rehabilitation Unit, University of Miyazaki Hospital, Miyazaki, Japan
[2] Department of Mathematics Education, Faculty of Education, University of Miyazaki, Miyazaki, Japan
[3] Department of Nursing, Faculty of Human Health Sciences, Shunan University, Shunan, Yamaguchi, Japan
[4] School of Nursing, Faculty of Medicine, University of Miyazaki, Miyazaki, Japan
[5] Center for Language and Cultural Studies, Lecturer, University of Miyazaki, Miyazaki, Japan
[6] Department of Orthopaedic Surgery, Faculty of Medicine, University of Miyazaki, Miyazaki, Japan

Corresponding author
Shigeaki Miyazaki,
03-5-23@med.miyazaki-u.ac.jp

## ABSTRACT

**Background**. Understanding the gait pattern of patients eligible for total hip arthroplasty (THA) due to hip osteoarthritis (OA) offers valuable information for improving locomotive syndrome (LS). This study aims to measure the gait patterns of THA-eligible patients using an optical motion capture system and to analyze these patterns using principal component analysis (PCA). Additionally, this study examines the relationship between THA-induced gait patterns and LS.

**Methods**. This before-after study included 237 patients who underwent unilateral primary THA due to hip OA. The primary outcome measures were spatiotemporal gait parameters. Secondary outcome measures included three LS risk tests: a stand-up test, a two-step test, a 25-question Geriatric Locomotive Function Scale (GLFS-25), and total clinical decision limits stages. PCA was performed using 16 spatiotemporal gait parameters collected before and three months after THA. Principal components (PC) were selected to achieve a cumulative contribution rate of 90% (0.9) or higher. Each summarized PC was compared using a paired $t$-test before and three months after THA. Furthermore, multiple regression analysis was conducted to determine how changes in each PC between before and three months after THA related to changes in the four LS evaluation items.

**Results**. PCA identified three principal components (PC1, PC2, PC3) that accounted for a cumulative contribution rate of 0.910 using 16 spatiotemporal gait parameters. When comparing before and three months after THA for all three PCs, significant differences were observed in each PC ($p < 0.001$), with overall walking ability and stance phase being higher three months after THA than before THA, while the asymmetry of support time was lower three months after THA. The results of multiple regression analysis revealed that PC1, PC2, and PC3 were the most influential factors in total clinical decision limits stage. For each LS risk test, the factors related to the stand-up

test were identified as PC1, PC2, and PC3, while the factors related to the two-step test were identified as PC1 and PC2. The factors related to the GLFS-25 were also identified as PC1 and PC2.

**Conclusions**. The most important findings of this study indicate that the three PCs represent over 90% of the 16 spatiotemporal gait parameters, which are associated with total clinical decision limits stage and LS risk tests. The present results suggest that PC1 represents overall walking ability, PC2 represents the stance phase, and PC3 represents asymmetry of support time. Gait pattern characteristics, such as overall walking ability, stance phase, and asymmetry of support time, were clearly defined by these PCs. Regarding the relationship between PC and LS, all three PCs are related to total clinical decision limits stage. In addition, PC1 and PC2 related to all three LS risk tests, and PC3 related only to the stand-up test.

# INTRODUCTION

Hip osteoarthritis (OA) is a degenerative joint disease characterized by clinical symptoms such as pain, limited joint range of motion, leg length inequality, and walking impairment. When conservative treatments for hip OA are ineffective, surgical intervention becomes necessary. Total hip arthroplasty (THA) is a beneficial surgical treatment for reconstructing hip function and improving pain and limitation of activities of daily living (ADL) in patients with end-stage OA. Previous studies have shown that the demand for THA is steadily increasing (*Ackerman et al., 2017*; *Kurtz et al., 2005*). Future demand growth rates for primary THA are estimated to be 174% from 2005 to 2030 in the United States (*Kurtz et al., 2007*), 134% from 2012 to 2030 in England and Wales (*Patel et al., 2015*), and 208% from 2013 to 2030 in Australia (*Ackerman et al., 2019*). Moreover, the demand for THA is expected to increase in many other countries.

The therapeutic effects of THA on daily living function have been demonstrated to improve walking function (*Laupacis et al., 1993*), sports activities (*Huch et al., 2005*), and cardiopulmonary function (*Ries et al., 1997*), among others. In particular, THA is reported to improve walking function in a short period of time (*Miyazaki et al., 2022*). In a recent systematic review, the biomechanical changes in gait before and after THA for hip OA were compared. The results showed relatively consistent improvements in walking speed, step length, and stride length (*Bahl et al., 2018*). However, abnormal gait patterns referred to as limping, such as trendelenburg gait and coxalgic gait, have been reported in patients with hip OA (*Lim et al., 2007*; *Trendelenburg, 1998*). Limping may persist after THA and can affect clinical outcomes and patient satisfaction (*Bonnefoy-Mazure et al., 2022*; *Pongcharoen & Chaichubut, 2019*; *Amlie et al., 2014*). Various factors contribute to walking abnormalities in patients eligible for THA in hip OA, including weakness in the abductor muscle groups (trendelenburg gait), attempt efforts to reduce joint forces due to pain (coxalgic gait), or

a combination of both (*Kim et al., 2024*; *Diamond et al., 2018*). In the analysis of gait in patients eligible for THA, it is common to measure spatiotemporal parameters, kinematic data, and kinetic data. Many studies have evaluated gait using these parameters. However, because different studies measure different spatiotemporal gait parameters, comparing results across studies is challenging (*Ewen et al., 2012*). To address this issue, this study examines which spatiotemporal gait parameters are most appropriate for use in gait analysis of patients eligible for THA.

Locomotive syndrome (LS) was introduced by the Japanese Orthopaedic Association (JOA) in 2007. LS refers to the decline in mobility due to musculoskeletal impairments hindering independence in daily activities (*Nakamura, 2009*). Multiple factors are associated with LS, including osteoarthritis, osteoporosis, osteoarthritis of the spine, spinal stenosis, and sarcopenia. As LS progresses, the risk of requiring nursing care increases (*Ishibashi, 2018*; *Nakamura & Ogata, 2016*; *Nakamura, 2008*). Mobility function is assessed by three LS risk tests: stand-up test, two-step test, and 25-Question Geriatric Locomotive Function Scale (GLFS-25) (*Ogata et al., 2015*). In addition, clinical decision limits are used as criteria to determine the risk of LS (*Yoshimura et al., 2015*). The clinical decision limits was introduced in 2015 from a two-stage system classified into stage 1 or stage 2 (*Ogata et al., 2015*), and stage 3 was added in 2020, resulting in a three-stage evaluation (*Miyazaki et al., 2021*). Since introducing the LS concept, studies have reported the effect of THA on gait improvement (*Miyazaki et al., 2022*; *Bahl et al., 2018*). However, there is no research examining the relationship between LS and gait characteristics in patients eligible for THA due to hip OA. Thus, in this study, we hypothesized that the gait characteristics of patients eligible for THA are related to the improvement of LS. The primary objective of this study is to measure the walking of patients with THA due to hip OA before and three months after THA and then identify the gait pattern characteristics using principal component analysis (PCA). The secondary objective is to examine the relationship of gait characteristics by THA on LS.

PCA is a statistical method for reducing many correlated original variables to their essential features, called principal components. Principal components are a few linear combinations of the original variables that maximally explain the variance of all the variables. For this reason, it has attracted interest in biomechanics (*Kobayashi et al., 2016*; *Federolf, Boyer & Andriacchi, 2013*; *Deluzio & Astephen, 2007*). Therefore, we attempted to aggregate numerous spatiotemporal gait parameters into fewer parameters using PCA. This approach provides health professionals with the knowledge to understand the gait characteristics of patients eligible for THA, enabling appropriate gait evaluation.

## MATERIALS & METHODS

### Study design and ethical statement

This before-after study was approved by the Research Ethics Committee of the University of Miyazaki, School of Medicine (Approval No. O-0783). It was carried out in compliance with the Ethical Guidelines for Medical and Biological Research Involving Human Subjects at the Department of Rehabilitation Medicine, University of Miyazaki Affiliated Hospital.

**Table 1  Subjects' characteristics.**

| Characteristic | Total ($n = 237$) |
| --- | --- |
| Age (y) | $67.7 \pm 9.2$ |
| Sex (male) | 46 |
| Height (cm) | $154.0 \pm 7.6$ |
| Weight (kg) | $58.5 \pm 11.1$ |
| BMI (kg/m$^2$) | $24.6 \pm 3.9$ |

Notes.
BMI, body mass index.
Age, height, weight and BMI values are means $\pm$ standard deviation.

Information regarding the conduct of the research including the objectives was disclosed and the research participants were provided an opportunity to refuse inclusion in the research.

## Patient selection

The participants of this study were patients who underwent unilateral primary THA due to hip OA between October 2018 and June 2023. Among them, patients who agreed to participate in all evaluations before and three months after THA were selected. Patients with osteonecrosis of the femoral head, trauma, rheumatoid arthritis, infection, revision THA, and those with incomplete outcome measure data sets were excluded from the study. After careful selection and rigorous screening, 237 patients were the participants of this study (Table 1 and Fig. 1). The distribution of total clinical decision limits stage was as follows: stage 1 (0 patients), stage 2 (20 patients), and stage 3 (217 patients). The cohort in this study included patients with polyarticular disease and those who have undergone arthroplasty in other joints.

All patients underwent THA using either an anterior minimally invasive surgical approach or a transgluteal approach performed by experienced surgeons at the hospital where the authors are affiliated. The standard length of hospitalization was 19 days. Rehabilitation days were 16 days, excluding the day of admission, the day of surgery, and the day of discharge. Post-THA rehabilitation began the day after surgery. During hospitalization, patients received 40 min of one-on-one rehabilitation twice a day. Throughout the outpatient period, patients received one-on-one rehabilitation for 20 to 40 min two to three times a week. The objectives for patients in three months after THA were to reduce pain, increase range of motion, achieve regular neuromuscular coordination, and improve walking patterns and activities of daily life (ADL).

## Outcome measures

The purpose of this study was to determine the gait characteristics before and three months after THA in patients with hip OA and to examine the impact of THA on LS. Therefore, the primary outcome measures in this before-after study were spatiotemporal gait parameters. Secondary outcome measures included three LS risk tests: stand-up test, two-step test, GLFS-25, and total clinical decision limits stage (Ogata et al., 2015). All measures were conducted before and three months after THA without using canes and crutches.

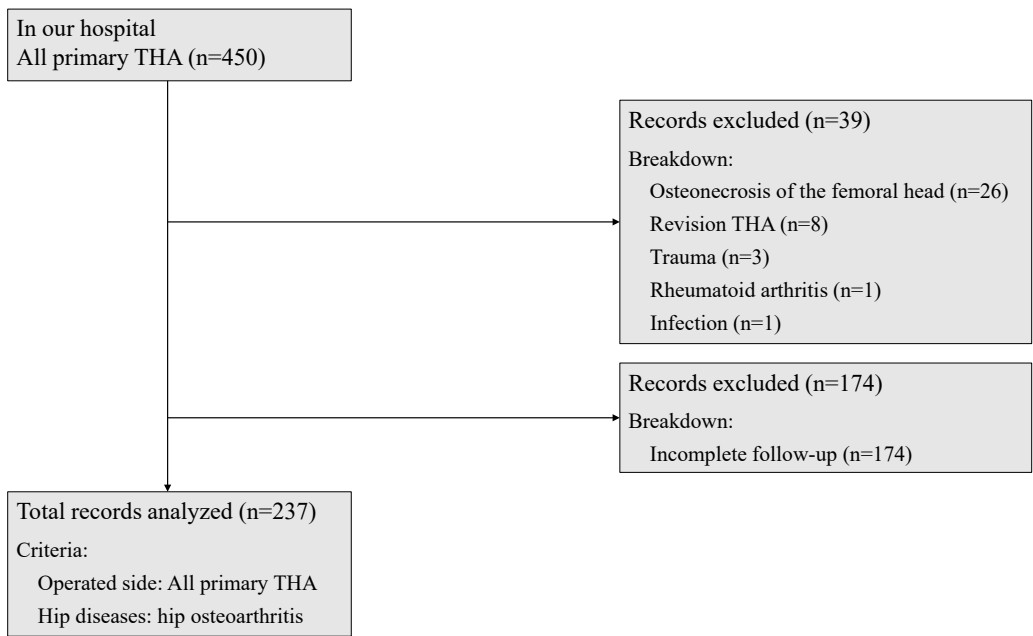

**Figure 1 Patient selection flow chart.** The subjects of this study were patients who underwent primary THA on the operative side for the hip OA. Patients with osteonecrosis of the femoral head, trauma, rheumatoid arthritis, infection, revision THA, and patients with incomplete outcome measure data sets were excluded.

### Primary outcome measures

Gait movements were measured using an optical motion capture (OMC) system (Vicon Nexus 2.12; Vicon Motion Systems, London, UK) equipped with 12 infrared cameras (Vantage 5 and Vantage 8; Vicon Motion Systems, London, UK) and six force plates (OR6-5 and BP400600; Advanced Mechanical Technology, Inc.,Watertown, MA, USA). The participants wore skin-tight clothing (Under Armour, Baltimore, MD, USA) with 35 reflective markers (14 mm diameter) attached according to the Plug-in-Gait model protocol (*Kadaba, Ramakrishnan & Wootten, 1990*). The sampling frequency was set at 100 Hz, and all the equipment took measurements synchronously. In this study, the OMC system was used to obtain spatiotemporal gait parameters. However, since this study does not use kinematic or kinetic data, any alternative systems capable of obtaining spatiotemporal gait parameters would be sufficient, and it is not necessary to use the OMC system.

Spatiotemporal gait parameter: The patient walked barefoot freely and was measured three times after practicing walking without discomfort. The data of the movement the patient was most satisfied with were adopted. An analysis software (Vicon Nexus 2.12) was used to calculate spatiotemporal gait parameters (cadence, stride time, step time, single support time, double support time, stride length, step length, and walking speed). First, the "Detect Events from a Bertec Force Plate Treadmill" program in Vicon Nexus was used to determine a gait cycle based on the timing of initial contact and toe-off. Then, the "Gait Cycle Parameters Calculator" program was used to calculate spatiotemporal gait

parameters based on the coordinate data of the reflective markers. The instructions for quantifying the movements were: (1) walk freely; (2) rest for 20 s between movements.

### Secondary outcome measures

LS was evaluated using the following three LS risk tests. The clinical decision limits and total clinical decision limits of each test were classified as stages 0–3. Measurement procedures and assessments were performed before and three months after THA according to the JOA guidelines. Data were collected as previously described in *Miyazaki et al. (2021)*.

1. The stand-up test: The stand-up test evaluates leg strength by the height at which the participant subject can stand up from both legs or a single leg at four different heights: 40 cm, 30 cm, 20 cm, and 10 cm (*Nakamura & Ogata, 2016*). The test starts with sitting on a 40-cm platform and standing up with both legs. If it can be successfully done, the procedure is repeated with single-leg standing alternating between the left and right legs. If the participant subject cannot stand up on either leg on the 40-cm platform, the test is considered a failure and repeated on a 10-cm lower platform. The scoring system employed nine performance scores (*Ogata et al., 2015*): 0 (inability to stand); 1, 2, 3, or 4 (stand using both legs from a height of 40, 30, 20, and 10 cm, respectively); and 5, 6, 7, and 8 (stand using one leg from a height of 40, 30, 20, and 10 cm, respectively). Clinical decision limits stages refer to the following: stage 3-unable to stand on both legs from a height of 30 cm; stage 2-unable to stand on both legs from a height of 20 cm but able to stand from a height of 30 cm; and stage 1-unable to stand on either leg from a height of 40 cm but able to stand on both legs from a height of 20 cm. Scores <2, <3, and <5 were classified as clinical decision limits stages 3, 2, and 1, respectively. The instructions to quantify motion were given. (1) Fold arms in front of the chest at a sitting position. (2) Place feet shoulder-width apart. (3) Position lower legs at an angle of approximately 70° to the floor. (4) Stand up without gaining momentum. (5) Maintain the standing posture for 3 s.

2. The two-step test: The two-step test evaluates walking ability, including lower limb muscle strength, balance, and flexibility (*Yoshimura et al., 2015*), by using the maximum two stride lengths that can be taken from a stationary standing position without losing balance. The test score was calculated as the "2-step value" by dividing the maximum two stride lengths (cm) by the patient's height (cm). The 2-step values <0.9, ≥1.1 to <1.3, and ≥0.9 to <1.1 were classified as clinical decision limits stages 3, 2, and 1, respectively. The instructions to quantify motion were given. (1) Align toes at the starting line with a stationary standing position. (2) Take two of the longest possible steps forward, then align toes together in a stationary standing position. (3) If the participant subject loses balance, start over. (4) Measure the stride length of the two steps. (5) Perform two times and adopt the better score.

3. GLFS-25 (*Seichi et al., 2012*): The GLFS-25 is an easy-to-use instrument for early diagnosis developed to screen the elderly with risk factors of motor dysfunction. The test is a self-administered questionnaire consisting of 25 questions: four questions on pain, 16 questions on ADL, three questions on social functions, and two questions on mental health in the past month. The 25 questions were graded on a 5-point scale from

0 (average) to 4 (most severe). The most severe was scored as 100 points, with higher scores indicating a decline in motor function. Clinical decision limits stage 3 scores ≥24 points, stage 2 scores ≥16 to <24 points, and stage 1 scores ≥7 to <16 points.

4. The total clinical decision limits: Total clinical decision limits stage is determined based on the results of the stand-up test, the two-step test, and the GLFS-25. Total clinical decision limits stage refers to the following: stage 3-progressive decline in mobility functions, which interferes with social participation; stage 2-progressive decline in mobility functions; and stage 1-beginning decline in mobility.

## Statistical analysis

To determine the gait characteristics of THA patients, PCA was performed using 16 spatiotemporal gait parameters before and three months after THA. Principal components (PC) were selected to achieve a cumulative contribution rate of 90% (0.9) or higher, aiming to capture as many gait characteristics as possible. Each PC summarized by PCA was compared using a paired $t$-test before and three months after THA. Multiple regression analysis was also performed to clarify how changes in each PC before and three months after THA related to the four LS assessment tests (stand-up test, two-step test, GLFS-25, and total clinical decision limits stage).

The objective of this study is to analyze the impact of spatiotemporal gait characteristics on LS. However, at the planning stage, no information on spatiotemporal gait characteristics was available, so the sample size was determined based on the improvement rate of total clinical decision limits before and after THA. According to *Miyazaki et al. (2022)*, the improvement rate of total clinical decision limits at three months after THA was estimated to be 46.7%. Assuming an improvement rate of 46.7%, the sample size required to achieve a statistical power of 90% or more, with the improvement rate of total clinical decision limits being significantly higher than 30%, was calculated to be 187. The hospital where the authors are affiliated collects approximately 40 participants per year. Therefore, a five-year data collection period was needed to reach a sample size of 187 or more. Consequently, the sample size was 237, resulting in a statistical power of more than 95%. All statistical analyses were performed using IBM SPSS 27.0 (2020; IBM Corp., Armonk, NY, USA). The statistical significance was set at $p < .05$.

## RESULTS

Table 1 shows the demographic data: age, sex, height, weight, and BMI. The participants were 237 patients comprising 46 males and 191 females, and the mean age was 67.7 years (standard deviation SD: 9.2 years). The mean BMI was 24.6 kg/m2 (standard deviation SD: 3.9 kg/m2).

The results of PCA using 16 spatiotemporal gait parameters (Table 2) showed that three components were extracted: PC1 had an eigenvalue of 9.191 (factor contribution rate of 0.574), PC2 had an eigenvalue of 4.494 (factor contribution rate of 0.281), and PC3 had an eigenvalue of 0.871 (factor contribution rate of 0.055). The cumulative contribution rate for the three PCs was 0.910. In the first principal component (PC1), the factor loadings had almost equal values, with absolute values between 0.2 and 0.3 for all spatiotemporal gait

**Table 2   Loadings of principal components ($n = 237$).**

| Spatiotemporal Gait Parameters | PC 1 | PC 2 | PC 3 |
|---|---|---|---|
| THAS Cadence | 0.266 | −0.244 | 0.016 |
| NTHAS Cadence | 0.269 | −0.238 | 0.047 |
| THAS Stride Time | −0.287 | 0.210 | 0.070 |
| NTHAS Stride Time | −0.287 | 0.208 | 0.043 |
| THAS to NTHAS Step Time | −0.283 | 0.158 | 0.285 |
| NTHAS to THAS Step Time | −0.247 | 0.239 | −0.224 |
| THAS Single Support Time | 0.000 | 0.353 | −0.665 |
| NTHAS Single Support Time | −0.139 | 0.333 | −0.020 |
| THAS to NTHAS Double Support Time | −0.291 | −0.016 | 0.347 |
| NTHAS to THAS Double Support Time | −0.298 | −0.047 | 0.347 |
| THAS Stride Length | 0.221 | 0.334 | 0.181 |
| NTHAS Stride Length | 0.224 | 0.331 | 0.104 |
| THAS to NTHAS Step Length | 0.202 | 0.318 | 0.207 |
| NTHAS to THAS Step Length | 0.211 | 0.303 | 0.112 |
| THAS Walking Speed | 0.293 | 0.176 | 0.220 |
| NTHAS Walking Speed | 0.295 | 0.177 | 0.170 |
| Eigen value | 9.191 | 4.494 | 0.871 |
| Proportion of variance explained | 0.574 | 0.281 | 0.055 |

Notes.

PC, principal component; THAS, total hip arthroplasty side; NTHAS, non total hip arthroplasty side.
Three components that reached a cumulative contribution rate of 90% were adopted.

parameters except THAS single support time and NTHAS single support time. Specifically, cadence, stride length, step length, and walking speed on both the THA side and non-THA side had positive values, while stride time, step time, and double support time, which are time-related parameters, had negative values. In the second principal component (PC2), 12 of the spatiotemporal gait parameters had positive values, with single support time, stride length, and step length on both the THA side and non-THA side having factor loadings of 0.3 or higher. The spatiotemporal gait parameters with negative values were cadence and double support time on both the THA side and non-THA side. In the third principal component (PC3), the factor loading for THA side single support time was the highest in absolute value at −0.665, while the two double support times had factor loadings of 0.347. The factor loadings for the other spatiotemporal gait parameters were 0.3 or less in absolute value.

When the three PCs summarized by PCA were compared before and three months after THA with a corresponding $t$-test, significant differences were found in PC1 ($t = 8.38$, $p < 0.001$), PC2 ($t = 4.61$, $p < 0.001$), and PC3 ($t = -3.21$, $p < 0.01$). The overall walking ability and stance phase were higher three months after THA than before THA, while the asymmetry of support time was lower at three months after THA (Fig. 2).

Multiple regression analysis revealed that PC1, PC2, and PC3 were the most influential factors related to total clinical decision limits stage, and their standard partial regression coefficients were −0.188 ($p < 0.01$), −0.129 ($p < 0.5$), and −0.218 ($p < 0.01$), respectively. For each LS risk test, PC1, PC2, and PC3 were identified as factors related to the stand-up

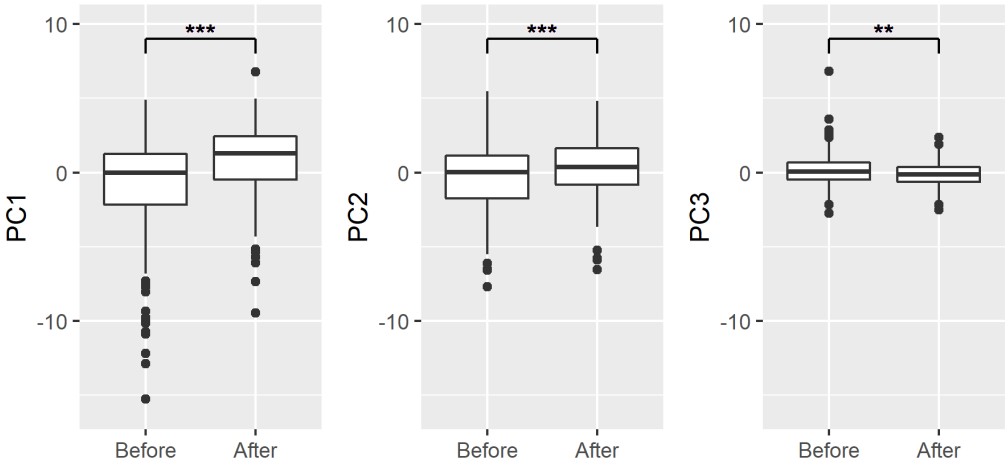

**Figure 2  Changes in each PC before and after THA.** Solid lines in the boxes indicate median values. Asterisk denotes statistically significant differences between the means using paired $t$-test (**$p < 0.01$, ***$p < 0.001$).

test, with standard partial regression coefficients of 0.318 ($p < 0.001$), 0.189 ($p < 0.01$), and 0.174 ($p < 0.01$), respectively. PC1 and PC2 were extracted as factors related to the two-step test, with standard partial regression coefficients of 0.510 ($p < 0.001$) and 0.321 ($p < 0.001$), respectively. PC1 and PC2 were extracted as factors related to GLFS-25, with standard partial regression coefficients of $-0.495$ ($p < 0.001$) and $-0.141$ ($p < 0.05$), respectively (Table 3).

## DISCUSSION

In this study, the participants were patients who underwent unilateral primary THA due to hip OA. OMC system was used to evaluate spatiotemporal gait parameters, three LS risk tests, and total clinical decision limits stage before and three months after THA. PCA was used to examine the gait characteristics in patients. The impact of gait patterns due to THA on LS was analyzed. According to the reports by systematic reviews of two- or three-dimensional gait analysis of THA patients, most sample sizes ranged from 10–30 patients, with a maximum of 145 patients (*Bahl et al., 2018*). To our knowledge, no study has utilized OMC system to analyze gait in such a large sample size of more than 200 THA-eligible patients due to hip OA, examined gait characteristics using PCA, or evaluated the relationship between gait characteristics in THA and LS. It increases the generalizability of the findings, providing a more comprehensive understanding of gait characteristics and their impact on LS in the larger population of THA patients. There were 22 spatiotemporal gait parameters generated by OMC system, six of which were secondary spatiotemporal gait parameters from other spatiotemporal gait parameters. Therefore, 16 spatiotemporal gait parameters were adopted, excluding these six in this study. The most important findings of this study were that three PCs could represent more than 90% of the information from

**Table 3  Results of multiple regression analysis for changes in locomotive syndrome ($n = 237$).**

| Dependent variables | Independent variables | Partial regression coefficients | SE | Standard partial regression coefficients | t | p |
|---|---|---|---|---|---|---|
| Stand-up test | PC1 | 0.100 | 0.020 | 0.318 | 5.004 | 0.000[***] |
| | PC2 | 0.094 | 0.031 | 0.189 | 3.043 | 0.003[**] |
| | PC3 | 0.141 | 0.051 | 0.174 | 2.779 | 0.006[**] |
| Two-step test | PC1 | 0.032 | 0.004 | 0.510 | 9.147 | 0.000[***] |
| | PC2 | 0.032 | 0.005 | 0.321 | 5.883 | 0.000[***] |
| | PC3 | −0.013 | 0.009 | −0.081 | −1.478 | 0.141 |
| GLFS-25 | PC1 | −2.634 | 0.317 | −0.495 | −8.301 | 0.000[***] |
| | PC2 | −1.181 | 0.489 | −0.141 | −2.413 | 0.017[*] |
| | PC3 | −1.309 | 0.803 | −0.096 | −1.630 | 0.105 |
| Total clinical decision limits stage | PC1 | −0.046 | 0.016 | −0.188 | −2.874 | 0.004[**] |
| | PC2 | −0.050 | 0.025 | −0.129 | −2.020 | 0.044[*] |
| | PC3 | −0.139 | 0.041 | −0.218 | −3.391 | 0.001[**] |

**Notes.**
SE, standard error; GLFS-25, 25-Question Geriatric Locomotive Function Scale.
Significantly different: *** $p < .001$, ** $p < .01$, * $p < .05$.
All variables were post-THA values minus pre-THA values.
Each score of stand-up test, two-step test, GLFS-25, and the stage value of total clinical decision limits stage was used.

16 spatiotemporal gait parameters, and the relationship between the three PCs and the commonly used LS risk tests was clarified.

## Characteristics of each PC and impact by THA
### PC1: Overall walking ability
In PC1, among the 16 spatiotemporal gait parameters, the absolute factor loadings of 14 variables were between 0.2 and 0.3. The variables with positive factor loadings included eight parameters: cadence, stride length, step length, and walking speed on both the THA and non-THA sides, indicating that higher values represent greater walking ability. On the other hand, the variables with negative factor loadings included six parameters: stride time, step time, and double support time on both the THA and non-THA sides, indicating that lower values represent greater walking ability. Therefore, PC1 was interpreted as representing the overall walking ability of THA patients, and the term "overall walking ability" was used. Comparing before and three months after THA, PC1 showed a significant change toward positive, indicating improved overall walking ability three months after THA.

### PC2: Stance phase
In PC2, the six variables with factor loadings of 0.3 or higher were single support time, stride length, and step length on both the THA and non-THA sides. A gait cycle is defined as "the period from the initial contact of one foot to the next initial contact of the same foot". The gait cycle is divided into two phases: stance phase and swing phase. The stance phase refers to the period when the foot is in contact with the ground, and all six variables

indicated the stance phase. Therefore, PC2 was interpreted as representing the stance phase of THA patients, and the term "stance phase" was used.

According to systematic reviews by *Bahl et al. (2018)*, the evidence for improvements in pre-THA compared with three months after THA showed that the prolongation of single support time was strong, the increase in stride length was moderate, and the increase in step length was strong. Furthermore, *Da Cunha et al. (2016)* also reported significant improvements in the prolongation of single support time and increases in stride length and step length in post-THA. From these findings, the stance phase of the gait cycle is particularly characteristic of THA patients, and the change toward positive in PC2 indicates improvement in gait function.

### PC3: Asymmetry of support time

In PC3, all variables with factor loadings of 0.3 or higher were related to support time. The double support time and single support time had opposite signs, indicating that higher values represent a shorter single support time on the THA side. The term "asymmetry of support time" was used to describe the differences between double support time and single support time on the THA side. In patients with limping, a shortened single support time on the THA side was observed compared to double support time on both the THA and non-THA sides, which can be inferred from the positive values of PC3.

Limping after THA is caused by pain, leg length inequality, nerve damage, and periprosthetic muscle dysfunction (*Horstmann et al., 2013*; *Röder et al., 2012*; *Khan & Knowles, 2007*). Additionally, limping is an adverse clinical outcome that affects patient satisfaction (*Pongcharoen & Chaichubut, 2019*). According to a large-scale cohort study that evaluated limping at the time of THA and five years after THA, the presence of limping after THA leads to patient dissatisfaction, and the strength of this association varies depending on the severity of the limping and the presence of associated pain (*Bonnefoy-Mazure et al., 2022*). PC3 is a component that evaluates the state of limping, and it is considered an important assessment indicator for improving patient satisfaction after THA.

## Impact of principal components changes on LS changes

It was reported that in patients with total clinical decision limits stage 3 before THA evaluation, significant improvements were observed in total clinical decision limits stage and all three LS risk tests after THA (*Miyazaki et al., 2022*). This study examined the gait characteristics that influence total clinical decision limits stage and all three LS risk tests. The results indicated that all PC1-PC3 significantly influenced total clinical decision limits stage and risk tests. Moreover, PC1 and PC2 were particularly influential on all three LS risk tests, while PC3 influenced on the stand-up test.

### Relationship between PC1, 2 and LS

PC1 demonstrated a strong influence on all three LS risk tests, particularly on GLFS-25. GLFS-25 is a self-administered questionnaire consisting of 25 questions designed as a screening tool for the elderly with mobility impairment, reflecting difficulties in ADL over the past month (*Akai et al., 2016*; *Seichi et al., 2012*). Regarding the relationship between motor function and GLFS-25, previous studies reported clinical outcomes and findings

of hip muscle strength three months after THA using GLFS-25 (*Maezawa et al., 2018*). However, there are no previous studies that have investigated the relationship between gait function and GLFS-25. The findings of this study suggest that improvements in overall physical walking ability strongly influence improvements in the GLFS-25 score obtained from the self-administered questionnaire. This implies that THA and rehabilitation have a significant impact not only on the improvement of overall physical walking ability but also on the improvement in ADL and social participation.

PC2 reveals a lesser influence than PC1 on all three LS risk tests. However, interestingly, it demonstrated an impact on the two-step test, which is used to assess walking ability. In a previous study *Kobayashi & Ogata (2018)*, PCA was used to examine the relationship between gait patterns during normal walking and the two-step scores. It was reported that elderly with lower two-step scores had a smaller range of motion in the sagittal plane of the hip, knee, and ankle joints during the stance phase. The correlation between the change in PC2 and the change in the two-step score suggests that the two-step score decreases as both stride length and step length decrease. This finding indicates that the change toward negative in PC2 signifies decreased movement of the lower limb in the sagittal plane during the stance phase, indicating that the stance phase is closely related to the two-step test.

### Relationship between PC3 and LS

Unlike PC1 and 2, PC3 did not demonstrate any improvements in LS. The results indicated that PC3 showed changes toward the opposite direction of improvement in regard to total clinical decision limits stage and stand-up test. Standing up requires lower limb muscle strength, joint range of motion, flexibility, and balance (*Nakamura & Ogata, 2016*; *Ogata et al., 2015*). Particularly, the range of motion of the hip and knee joints and knee extension muscle strength in the quadriceps femoris muscle are important. The outcomes of the present study indicate that PC1 and 2 had significant influences, while PC3 had minimal impact. PC3 showed a change toward negative, which means an improvement in limping three months after THA. Unfortunately, the improvement in limping did not lead to an improvement in LS. This may be due to the fact that only a limited number of participants in this study had strong limping before THA.

## CONCLUSIONS

This study targeted patients who underwent unilateral primary THA, and PCA of gait characteristics were examined. The results indicated that three PCs accounted for more than 90% of 16 spatiotemporal gait parameters. PC1 represented overall walking ability, PC2 characterized the stance phase, and PC3 depicted the asymmetry of support time. Both PC2 and PC3 were identified to represent the gait characteristics of THA. Regarding the relationship between PCs and LS, all three components influenced the total clinical decision limits stage. Specifically, PC1 and PC2 were correlated with all three LS risk tests, whereas PC3 was associated primarily with the stand-up test. These findings are clinically significant for the rehabilitation of THA due to hip OA, facilitating a more effective evaluation of motor function and ADL.

## ACKNOWLEDGEMENTS

The authors thank Masaru Ochiai, Aya Unoki, Wakaba Iha, and Akari Nagatomo for their assistance with the experimental measurements.

### Funding

This work was supported by the Japan Society for the Promotion of Science Grants-in-Aid for Scientific Research (Grant No. 24K20438). There was no additional external funding received for this study. The funders had no role in study design, data collection and analysis, decision to publish, or preparation of the manuscript.

### Grant Disclosures

The following grant information was disclosed by the authors:
Japan Society for the Promotion of Science Grants-in-Aid for Scientific Research: Grant No. 24K20438.

### Competing Interests

The authors declare there are no competing interests.

### Author Contributions

- Shigeaki Miyazaki conceived and designed the experiments, performed the experiments, analyzed the data, prepared figures and/or tables, authored or reviewed drafts of the article, and approved the final draft.
- Yoshinori Fujii conceived and designed the experiments, analyzed the data, prepared figures and/or tables, authored or reviewed drafts of the article, and approved the final draft.
- Kurumi Tsuruta conceived and designed the experiments, analyzed the data, prepared figures and/or tables, authored or reviewed drafts of the article, and approved the final draft.
- Saori Yoshinaga conceived and designed the experiments, analyzed the data, prepared figures and/or tables, authored or reviewed drafts of the article, and approved the final draft.
- Amy Hombu conceived and designed the experiments, analyzed the data, prepared figures and/or tables, authored or reviewed drafts of the article, and approved the final draft.
- Taro Funamoto performed the experiments, authored or reviewed drafts of the article, and approved the final draft.
- Takero Sakamoto performed the experiments, authored or reviewed drafts of the article, and approved the final draft.
- Takuya Tajima performed the experiments, authored or reviewed drafts of the article, and approved the final draft.

- Hideki Arakawa performed the experiments, authored or reviewed drafts of the article, and approved the final draft.
- Tsubasa Kawaguchi performed the experiments, authored or reviewed drafts of the article, and approved the final draft.
- Jun Nakatake performed the experiments, authored or reviewed drafts of the article, and approved the final draft.
- Etsuo Chosa conceived and designed the experiments, analyzed the data, authored or reviewed drafts of the article, and approved the final draft.

### Human Ethics

The following information was supplied relating to ethical approvals (*i.e.*, approving body and any reference numbers):

This study was approved by the Research Ethics Committee of the University of Miyazaki, School of Medicine (Approval No. O-0783).

### Data Availability

The raw measurements are available in the Supplemental File.

### Supplemental Information

Supplemental information for this article can be found online at http://dx.doi.org/10.7717/peerj.18351#supplemental-information.

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
