# Peer review of "Spatiotemporal gait characteristics post-total hip arthroplasty and its impact on locomotive syndrome: a before-after comparative study in hip osteoarthritis patients"

_PeerJ, doi:10.7717/peerj.18351_

## Round 0.1 · original submission · Major Revisions

We believe that your work should be considered for publication, but major revisions are required. Please revise according to the reviewer comments, and provide a point-by-point response to the comments.

I agree with reviewer 2 that the mention of motion capture is not consistent with the work that was actually performed. I also suggest to change the title, and remove the words "Three-dimensional analysis of gait characteristics". There is nothing three-dimensional about spatiotemporal parameters, and "gait characteristics" suggests something more general. You might consider using the term "spatiotemporal gait characteristics".

·

Basic reporting

The manuscript is very well written, and I have no concerns regarding the use of language; the manuscript is appropriately structured.
The introduction section provides a very strong literature review, addressing the prevalence of hip replacement surgery and the implications of arthritis and consequent surgery, the effects of arthritis and surgery on gait parameters, locomotive syndrome, and principal components analysis.
In line 88, the authors state that no standardized walking parameter exists for gait analysis. This comment puzzles me. Perhaps I do not understand what they mean by “standardized.” In fact, many commonly used gait parameters are routinely used in gait analysis, including almost all of those used by the authors in the current paper. The authors need to explain this statement.
The objectives of the study are clearly stated. However, no hypothesis statements are provided.
Starting in the paper's methods section, I appreciate that the word ‘patient’ is used, as the individuals were patients receiving treatment before and during the study. And I note that at various points in the manuscript the individuals are referred to as ‘subjects’. For the purposes of describing the study, however, I suggest that these individuals should be referred to as ‘participants.’ Within the context of this study, they are not patients but rather volunteer participants. That they are patients is no more or less relevant to the study than their age, sex, or any other descriptor. Further, I do not refer to participants as ‘subjects’ because, as researchers, we do not subject anyone to our methods (unlike animals such as mice and fish, which do not volunteer to participate but are subjected to our methods). I will leave it to the editor, but this is my perspective.

Experimental design

Generally, the experimental design is very good. The inclusion and exclusion criteria are clearly stated. The participant selection/screening methodology was robust.
Data collection methodologies were clearly described. I appreciate the multi-modal nature of the aggregated data set, which included spatiotemporal gait parameters, three different tests of locomotive syndrome, and the CDL metric.
On line 216, the authors implied causality by using the word “affected.” More appropriately, they should describe a (potential) relationship between PC and LS test changes. Changes in PCs and LS test outcomes may be related to one another, but it is impossible to suggest that changes in PCs caused changes in LS test outcomes.
In lines 218 to 220, the authors describe how they determined their sample size. The method or rationale they described is not an adequate method of determining sample size. The authors need to provide a proper sample size estimation calculation to determine an appropriate sample size to achieve an appropriate level of power (such as 0.8). I’m surprised the research ethics board did not require this calculation and determination before approving the protocol.
Lines 271 to 273, also regarding the sample size, the authors state that no other study has included 200 or more participants who are patients of the same nature as included in the current study and used motion capture to examine gait using PCA or evaluated the relationship between gait characteristics in THA and LS. This seems to be presented as a point of pride or differentiation. I think it’s wonderful that the authors can collect data with so many patients; many clinical biomechanists would be envious. However, the fact that they have collected data with over 200 participants when no other study has done so does not enhance the value of the study. In fact, it calls into question whether they subjected too many people, more than they needed to the experimental methods. This is a question of research ethics. I refer back to my previous comment regarding the need for a robust sample size estimation.

Validity of the findings

The interpretation of the data and the results is largely focused on the output of the principal components analysis. As such, the rest of my comments focus on the PCA.
In lines 276-278, the authors wrote, “The results suggest that PC1 represents overall walking ability, PC2 represents stance phase, and PC3 represents asymmetry of support time. Moreover, PC2 and PC3 reveal the characteristics of gait in THA.” On what basis are any of these statements made? I have examined the loading factors of each gait parameter to each of the three PCs that the authors have reported (table 2), and I don’t understand why the authors have suggested that the three PCs reflect these different characteristics (i.e., overall walking ability, stance phase, and asymmetry of support time). The authors must explain why (and how) they made these statements.
On what basis did the authors choose 90% as the threshold for the cumulative contribution? I note that the first two PCs together account for 85.5% and that PC3 provides only an arguably negligible contribution of 5.5%. Without a solid rationale for setting the cumulative contribution threshold at 90%, what rationale do the authors have for including PC3 when its contribution is so minor?
Because of these questions regarding the PCA outcomes, I cannot assess the validity of the rest of the analysis, which is related to the multiple regression.

Additional comments

I applaud the authors’ work. I am confident they can contribute substantially to the broader literature with this paper once they have addressed the various questions and concerns I have outlined.

·

Basic reporting

1. The authors present a study in which they identify the primary modes of variation in walking gait for people with total hip replacement. The authors run the risk of misleading the reader about the nature of their work when making reference to motion capture data. Motion capture data are used to compute spatiotemporal metrics. These could very easily be calculated from much simpler and cost efficient systems. What is present as a strength is in fact a potential weakness of the study.
2. Bahl et al is reported as 1997 in text. This is incorrect. This systematic review was published in 2018.
3. I strongly recommend against the use of non-standard acronyms as they dramatically reduce readability. I had to constantly go back to the original description of the term to ensure I correctly understood a sentence/table.
The reporting of the results is challenging to navigate. Specifically lines 231-247 was simply a reproduction of the information in the table.

Experimental design

1. The justification for the study lacks clarity. The introduction needs to better explain why the study is needed.
2. Please advise the tools used to calculate spatiotemporal parameters. Vicon Nexus is a capture tool.
3. Why did the authors use optical motion capture to spatiotemporal data? It seems as though this is an overly complex process.
4. There is no justification for the statement on line 88, i.e., "no studies have examined the gait patterns of more than 2000 THA…" . Why is 200 a relevant number? How can the authors be sure that previous studies weren't powered and that n=220 is? This is an especially important point as the authors come back to this on multiple occasions.
5. The technical description of the PCA lacks detail.

Validity of the findings

VALIDITY OF THE FINDINGS
1. The authors failed to report if the patients enrolled in the study had any degenerative changes / joint replacement in any of their other joints.
2. The variation in surgical approach is likely in itself to result in different early outcomes. There is extensive evidence showing that the direct anterior approach results in an accelerated pathway to recovery.
3. The authors should confirm their length of stay data. Most centres around the world aim to discharge within four days.
It is unsurprising that the authors identified that their PCA analysis explained ~60% of the variance in locomotor syndrome. The secondary outcomes have some element of locomotor function, but there are many additional facets to consider here. I wonder what value these results. The authors need to clarify this.

Additional comments

In summary, while it is clear that the authors have led a substantial program of work, I believe that there are three main factors that negatively impact the validity and potential impact of the results:
1) The patient cohort is inadequately described which means that any change in outcomes could be driven by surgical approach or secondary pathology.
2) The use of optical motion capture is inappropriate in this circumstance.
There is no justification for why we need larger samples sizes. While this is obvious on face value, the authors fail to truly justify their decisions.

---

## Round 0.2 · Minor Revisions

The reviewers provided some useful final comments. I believe that this will be a very nice publication after the remaining issues are addressed.

·

Basic reporting

The authors have addressed almost all of my earlier concerns.

Experimental design

The one remaining area of concern that I have is with respect to sample size. The authors might have misunderstood my point concerning sample size and the need for a power calculation. The authors responded that they could not estimate the number of patients available over the following five years. That is, they could not estimate how many patients would be available to participate in the study. My concern wasn’t related to how many would be AVAILABLE, but rather how many would be NEEDED to appropriately minimize the risk of a type 2 error. The sample they used was a convenience sample predicated on whomever was available. They collected data with 237 participants. How are we to be sure that that is a large enough sample size?
Furthermore, and of equal concern, how are the researchers to have known if they collected data with more participants than they needed? Perhaps they could have stopped with 100 patients and avoided the cost of collecting more data and as importantly, avoided the burden on 137 hip replacement patients. They need to provide a reasonable sample size estimation, which addresses the number of participants NEEDED for the study. To realize the importance of this matter, the authors might consider the question, “If I attempt to replicate the results of this study in my own lab, how many participants do I need to recruit?”

Validity of the findings

No comment

Additional comments

No comment

·

Basic reporting

Thank you for taking on board the comments I raised in my first review of this paper. I have some follow-up comments.

I don’t believe that the authors have made a compelling case for using optical motion capture for spatiotemporal gait parameters. There is no evidence provided. To build on my first review, we run the risk of others believing this is the best way to capture such data when it is very expensive and alternative systems exist with comparable resolution and accuracy (e.g., gaitrite mat). The authors need to consider what level of accuracy is required and put this into the context of the minimal clinically important differences for spatiotemporal gait parameters.

The authors should ensure that they refer to spatiotemporal gait parameters throughout to avoid misleading the reader. There are many occasions where the authors use the slightly nebulas term of gait parameters.

Experimental design

The authors will need to be very clear about how they used Vicon Nexus to computer parameters. The majority of the field will export the coordinates for analysis in third party software (e.g., matlab). The response provided by the authors to my original concern lacks any detail.

The response to my criticism regarding the lack of detail on other joint disease or arthroplasty is insufficient. I appreciate the aims of the study, but it is critical that any study accurately reports the cohort. At this stage, it is impossible to tell if the cohort is comprised of just single joint participants or if this is a normal clinical mix of people with multi-joint disease. Either outcome is fine. It’s just important to be transparent with reporting.

Validity of the findings

Refer to my comment on reporting the cohort characteristics.

Additional comments

I would like to thank the authors for clarifying the length of stay in Japan. While this is very different to many other regions it is helpful for context and good to reflect on.

---

## Round 0.3 · accepted · Accept

Thank you for addressing the remaining reviewer comments. The manuscript is now ready for publication.

·

Basic reporting

NA

Experimental design

NA

Validity of the findings

NA

Additional comments

The Authors have addressed my comments and present a nice paper.